# Internalization of the Western Standard of Beauty and Body Satisfaction: Evaluation Utilizing COPS and SATAQ-3 Questionnaires among Girls with Scoliosis

**DOI:** 10.3390/medicina59030581

**Published:** 2023-03-15

**Authors:** Jakub Glowacki, Joanna Latuszewska, Natalia Skowron, Ewa Misterska

**Affiliations:** 1Department of General Orthopaedics, Musculoskeletal Oncology and Trauma Surgery, Poznan University of Medical Sciences, 61-701 Poznan, Poland; 2The Faculty of Educational Studies, Kazimiera Milanowska College of Education and Therapy, 61-473 Poznan, Poland; 3Center for Early Intervention of the Polish Association for People with Intellectual Disabilities in Poznan, 61-446 Poznan, Poland; 4Department of Pedagogy and Psychology, University of Security, 60-778 Poznan, Poland

**Keywords:** scoliosis, body image, body dissatisfaction, psychometric properties

## Abstract

*Background and Objectives*: Patients with adolescent idiopathic scoliosis (AIS) more frequently present significant back-related body image disturbances compared with healthy controls. The study aimed to adapt two screening questionnaires: Sociocultural Attitudes Towards Appearance Questionnaire (SATAQ-3) and Cosmetic Procedure Screening Questionnaire (COPS), that could identify AIS patients, especially those threatened with body image disorders and might predict dissatisfaction with a desired-by-patients cosmetic result of treatment. *Materials and Methods*: In total, 34 AIS patients who undergo Cheneau brace treatment were asked to complete SATAQ-3 and COPS. *Results*: AIS patients presented a high level of internalization. Clinical and radiological factors that play a crucial role in the evaluation and decision process during brace treatment were not significantly associated with COPS and SATAQ-3 total scores. The SATAQ-3 total score and COPS results were also not related to sociodemographic parameters of the analyzed group. *Conclusions*: The presented study confirms the usefulness of the questionnaires, which aimed to isolate sociocultural risk factors of body image disorders in scoliosis patients as predictors of treatment dissatisfaction and worse compliance.

## 1. Introduction

Current research shows that sociocultural pressure to be thin and attractive could be the source of negative feelings about the body [1,2]. The internalization of media ideals and cultural pressure could quickly turn to body dissatisfaction concerns [3,4]. We must bear in mind that in the process of internalization, the person adopts their own certain behavior which previously appeared only as a direct reaction to environmental stimulation [5]. This adoption of body image created by media push women/men to body hyperawareness, leading to body shame and social anxiety [6,7].

Among patients’ motivations to undergo plastic surgery procedures, thus an improvement in their appearance, enhancements in self-confidence, self-esteem, and social interactions are the most common. The vast majority of patients are satisfied with cosmetic surgery. However, a specific group of patients is not [8]. Dissatisfaction can occur due to a preexisting psychiatric condition disregarding clinical outcomes [9].

Adolescent idiopathic scoliosis is a common skeleton deformity with 2–3% prevalence. A three-dimensional deformity occurs in frontal (lateral bend), sagittal (the deformation of thoracic kyphosis and lumbar lordosis), and transverse (associated with the rotation and translation of vertebrae) planes [10]. The progression of scoliosis is more prevalent in girls. As a result, girls need non-surgical or surgical treatment much more often than boys [11]. The progression of idiopathic scoliosis could manifest itself through multiple body deformities, including scapular and rib prominence, uneven shoulders, and an asymmetric waistline. According to several authors, those deformities can significantly influence self-esteem and body image [12,13,14]. Expressly, Auerbach et al. demonstrated more significant back-related body image disturbances in patients with scoliosis compared to healthy controls [8]. Spinal fusion surgery with instrumentation often reduces severe curves and minimizes the risk of curve progression [10,11]. Unfortunately, the technical success of the surgery does not correspond with patient satisfaction regarding surgical outcomes. Patient contentment with the surgical result appears to be unrelated to the medical aspects of spinal fusion surgery. This includes fundamental issues such as the preservation of pulmonary function and prevention of early osteoarthritis [15].

It should be emphasized that results concerning the body image assessment seem apparently contradictory. It was revealed that body dissatisfaction is related to the individual’s disfigured shape and weight. However, improving body shape and weight loss does not necessarily decrease body dissatisfaction [16]. This discrepancy might be explained by the Allocentric Lock Hypothesis (ALH) [17]. It implies that a basic disturbance may cause the maintenance of body dissatisfaction in the way the body is remembered [18]. Patients may be locked into remembering their body negatively, of which is not updated by contrasting self-centered representations driven by perception [19].

Returning to the main thread, it is still unclear under which circumstances patients with AIS revalue the perception of scoliosis-related body deformation. It is also unclear how those patients experience body image after operative treatment and how this overestimation is characterized. Those findings suggest that patients with AIS might need support while changing their desired body shape to feel more optimistic about the cosmetic results of scoliosis treatment [18,19].

A preoperative assessment tool that helps to identify patients with underlying psychiatric issues, improper motivations, or unrealizable expectations may help avoid such situations by providing a reason for psychological referral. However, it would not replace standard clinical examination but could alert the medical staff for possible psychological consultation.

The Sociocultural Attitudes Towards Appearance Questionnaire (SATAQ) measures the awareness of the cultural ideal of beauty for women and the level of acceptance and internalization of that ideal. The recent version, SATAQ-3, is one of the gold instruments used worldwide [20]. The SATAQ-3 has shown a relatively stable internal structure and boasts good indicators of reliability and validity when applied to Western women [20]. Another helpful questionnaire proven to be a valid, feasible, and reliable screening tool for patients is the Cosmetic Procedure Screening Questionnaire (COPS). It can be applied to cosmetic practice before performing any intervention. The scale has a high sensitivity for diagnosing Body Dysmorphic Disorder (BDD) in people likely to undergo a cosmetic procedure [21]. Therefore, our study aimed to adapt and analyze the psychometric properties of the Polish versions of COPS and SATAQ-3, which could identify AIS patients especially threatened with body image disorders and might predict dissatisfaction with a desired cosmetic result of brace treatment. The study represents a preliminary validation.

## 2. Materials and Methods

### 2.1. Study Design

This was a prospective study. Patients were purposely assigned to undergo the Cheneau brace intervention and were recruited from an academic center, where brace treatment was implemented.

The inclusion criteria were: female patients diagnosed with AIS with a Cobb’s angle between 20° and 40°, apical vertebra below T_8_, 10–14 years of age, and a Risser sign ≤2 at the beginning of brace treatment. The minimum follow-up of brace treatment was 3 months. According to Sponseller, patients with a high thoracic curve and a major curve apex above Th8 were excluded [22]—those patients could not be treated with the thoraco-lumbo-sarcal orthoses used in current study. The exclusion criteria were also mental disorders associated with developing a distorted body image (psychotic, dysmorphia). None of the patients had received any other spinal treatment before.

Concerning the bracing protocol, all patients were initially prescribed full-time bracing. The data regarding brace-wearing time compliance were based on anamnesis with patients during the clinical examination every three months. To verify patient claims concerning the time of bracing, an additional interview was conducted with the parents. Regarding the study protocol, all the participants and their parents received thorough information on the aim of the study. They were assured of anonymity, following which they gave informed consent. Then, they were asked to fill out the study questionnaires twice at home, with a 2-day interval, and return them by mail.

### 2.2. Study Sample

The study group consisted of 34 AIS females. Figure 1 presents the entire process of patient recruitment to the study and data collection. Thoracic scoliosis was identified in 55.8% of the patients (*n* = 19), thoracolumbar scoliosis in 26.47% (*n* = 9), and lumbar scoliosis in the remaining 17.64% (*n* = 6). The additional information about the study group is contained in Table 1.

### 2.3. Cultural Adaptation of Original Versions of COPS and SATAQ-3

We adopted the SATAQ-3 and COPS to Polish cultural settings. The process of cultural adaptation of the questionnaires was compliant with the guidelines of the International Quality of Life Assessment (IQOLA) Project and comprises the translation procedures and analysis of psychometric properties of adapted questionnaires [23].

#### 2.3.1. Questionnaires

The COPS screens for attribute(s) that the surveyed participants find unattractive, the characteristics of the cosmetic procedures, and the diagnostic criteria of BDD. Answers to questions are made by a Likert scale which helps the surveyed to quantify responses. The total score is calculated by summing questions from 2 to 10 (items 2, 3, and 5 are reversed). It ranges from 0 to 72 with a cut-off value of ≥40, indicating a high probability of having BDD [24]. Those individuals should be referred for further assessment. The scale is available online to download for free [25]. The aforementioned questionnaire is sensitive to changes and has good test–retest reliability, convergent validity, and strong internal consistency [24].

The SATAQ-3 is a 30-item scale with four theoretical subscales. The first subscale, with nine items, is Internalization-General and evaluates the impact of the general media related to TV, newspapers, and movies. The second subscale, with five items, is Internalization-Athlete and evaluates the internalization of gymnastic and sports models. The other subscales are Information, with nine items. It evaluates how far it is recognized that different media are considered significant sources of information about looks. The subscale Pressures, with seven items, evaluates individual feelings of pressure from display to media images and messages to modify one’s look [20]. The question format is a Likert-type scale ranging from 1 = completely disagree to 5 = completely agree. The total ranges from 30 to 150 with a cut-off value of ≥60, indicating a high probability of having BDD [20].

#### 2.3.2. Translation Procedure

The expert committee cooperated with the researchers to translate COPS and SATAQ-3, using the forward–backward method. At first, two experts working separately translated the questionnaire into Polish. Polish was the native language of these translators. In the second stage, these translations were synthesized into one version by translators and authors of the project. Then, two native English speakers who were bilingual back-translated COPS and SATAQ-3 into English. These translators did not know the original English version. Finally, all translators, a clinical psychologist, a statistician, and orthopedic surgeons reviewed all the translations. A consensus concerning all the inconsistencies was reached, establishing the Polish version of the SATAQ-3 and COPS. Finally, study participants filled in Polish versions of COPS and SATAQ-3 twice.

#### 2.3.3. Statistics

To determine the psychometric properties of Polish versions of COPS and SATAQ-3, statistical analysis was performed. Descriptive statistics (mean, 95% confidence intervals, range, and standard deviations) were utilized to describe the distribution of the results with respect to quantitative statistical features. With respect to qualitative features, we assigned percentages to the number of units belonging to the described categories of a given feature.

In addition, the psychometric properties of the COPS and SATAQ-3 were determined by the distribution of scores and determination of the floor effect (% of patients with the minimum score) and ceiling effect (% of patients with the maximum score).

Internal consistency was examined using Cronbach’s alpha coefficient. Cronbach’s alpha coefficient values were accepted as follows: excellent >0.80, adequate 0.70–0.79, and poor <0.70 [26]. To assess temporal stability, the test–retest method was used. The COPS and SATAQ-3, as pointed out above, were filled in twice and then the Intraclass Correlation Coefficient (ICC) was calculated. Discriminant validity was assessed by calculating the correlation between the total COPS and SATAQ-3, and the Cobb angle.

In addition, the responsiveness of the COPS and SATAQ-3 were determined by the means of effect sizes, which were calculated for each measure by dividing the mean absolute change score by the standard deviation of the baseline score. The interpretation of the magnitude of the effect size was based on Cohen’s rule-of-thumb, whereby an effect size of 0.2–0.5 was considered as small, 0.5–0.8 as moderate, and over 0.8 or greater represented a large effect.

Spearman’s rank-order correlation coefficients were used to evaluate the correlations between COPS and SATAQ-3 results and detailed the clinical and radiological characteristics of study participants, such as body mass index, age, Cobb angle, angle of trunk rotation, apical translation, and daily and monthly duration of brace wearing. Correlations were defined as strong—0.60, moderate—0.30–0.60, and weak—0.30, respectively.

We adopted *p* = 0.05 as the border level of statistical significance; test results with a *p*-value exceeding this level were considered insignificant. Statistical calculations were performed by the means of Statistica software.

### 2.4. Ethical Considerations

The study was conducted according to the guidelines of the Declaration of Helsinki and approved by the Ethics Committee (No. 1148/12).

## 3. Results

### 3.1. Outcome Measures–Scores Distribution

Minimal values, maximal values, means with SDs, and quartiles are presented in Table 2. Interestingly, a COPS total score ≥ 40 points, indicating a high level of BDD, applied to 5.88% of patients (one participant) in the test and 2.94% of patients in the retest (one participant). However, a SATAQ-3 total score of over 60 points, indicating a high level of Internalization, applied to 67.64% of patients in the test (23 participants) and 61.76% of patients in the retest (21 participants).

Table 3 presents the percentage of concern allocated to each feature according to COPS (from the pie chart regarding the 1st Feature to 5th Feature).

### 3.2. Psychometric Properties of SATAQ-3 and COPS

Floor and ceiling effects were present in the test and retest (see Table 4). To assess internal consistency, we used Cronbach’s alpha. Coefficient values equaled 0.79 (95% CI 0.67–0.88) and 0.93 (95% CI 0.89–0.96) for COPS and SATAQ-3, respectively. Similarly, temporal stability (test–retest reliability) based on the Intraclass Correlation Coefficient (ICC) was excellent and equaled 0.98 (95% CI 0.97–0.99) and 0.98 (95% CI 0.96–0.99) for COPS and SATAQ-3 total scores, respectively. These values are comparable with the psychometric properties of the original English version [20,24].

We also estimated the item–total correlation of both COPS and SATAQ-3 (see Table 5 and Table 6). Only one item in both questionnaires (item No. 1 for COPS and item No. 9 for SATAQ-3) was below the value of 0.20.

### 3.3. Correlation Analysis by Sociodemographic Characteristics

We assessed the relations between the sociodemographic characteristics of study participants. Age, place of residence, type of school, physical education, or recreational sports activity were not significantly associated with the SATAQ-3 total score. COPS results were also not related to sociodemographic parameters (for details, see Table 7).

### 3.4. Correlation Analysis by Clinical and Radiological Parameters

Interestingly, correlations between clinical and radiological parameters that play the primary role in the evaluation and decision process during brace treatment were not significantly associated with COPS and SATAQ-3 total scores (see Table 8).

### 3.5. The Logistic Regression Analyses

The multivariate model could not be applied due to the relatively small sample size (*n* = 34) for the SATAQ-3, and the small number of patients reporting high levels of BDD according to COPS.

## 4. Discussion

The research has demonstrated that body image is a significant concern among girls and women [27]. Many female adolescents attempt to have a perfect body appearance. Unfortunately, if they do not achieve their ideal appearance, they could develop negative emotions, leading to stress and anxiety [3]. Additionally, the mass media play a crucial role in distributing these body image criteria [3,28]. Thus, the psychometric assessment of body image could play a significant role in the preliminary selection of those who should be referred to cognitive-behavior therapy [29,30].

Referring to those assessment tools, Thompson et al. recommended the utilization of SATAQ-3 because of its excellent psychometric properties, the availability of its translated versions in different languages, and its structure dimensions [20]. However, many researchers use this scale mainly to study the relationship between sociocultural appearance standards promoted in the media and the selected risk factors concerning eating disorders [31]. The first Polish attempts to use SATAQ-3 took place in the research of Izydorczyk and Lizynczyk [32]. So far, there has been no Polish standardization of a tool for measuring body image and physical appearance standards promoted by mass media.

Additionally, COPS, recognized as a reliable screening tool, is sensitive to changes in patients under cognitive-behavior therapy. It may be used as an outcome measure after treatment to assess if there is any improvement in the symptoms of BDD [32].

However, it was found in our studies that in general, AIS patients perceived their appearance positively. These findings are similar to the results presented by Misterska et al. where as many as 60% of the study sample would not change anything [12]. Interestingly, a COPS total score ≥40 points, indicating a higher level of BDD, applied to 5.88% of participants and 2.94% in the retest, which is above general population estimation, but still significantly lower than in cosmetic surgery patients [29,30]. As previously stated, until recently, there were few studies that examined the perception of deformities in scoliosis patients [33]. This is probably because it is difficult to objectively assess the self-image perceptions of scoliotic patients.

Returning to the main thread, the percentage of concern allocated to each feature in the COPS questionnaire was mainly concentrated on scoliosis in general (35.29%) and its elements, such as rib prominence or uneven waist. Those findings are consistent with the results derived from the study performed by Misterska et al., finding that 29% of patients indicated rib prominence as the element of trunk deformity which is the most disturbing to them [14]. Interestingly, the COPS total score was correlated neither to radiological parameters nor to clinical data, such as age at diagnosis and brace time treatment. Therefore, according to Wilhelm et al., a thorough identification of those with a high probability of having BDD enables an early referral to a trained psychologist before attempting any medical intervention. Those people would most likely benefit more from cognitive-behavioral therapy [34].

The present research proves that the sociocultural appearance standards that are promoted in the mass media play an essential role in the studied group. In particular, the findings revealed that internalization of the Western standard of beauty and body satisfaction among adolescent girls with idiopathic scoliosis is neither correlated in regard to sociodemographics nor to clinical parameters. Moreover, our study revealed that radiological parameters, which play a crucial role in outcome measurements and potential further surgical treatment, are not correlated with COPS and SATAQ-3 scores. However, body image in scoliosis patients significantly affects appearance perception [14].

On the other hand, our study had limitations—we did not perform an objective trunk shape analysis. Only a standard, radiographic assessment was utilized. Meanwhile, we must bear in mind that radiographs do not provide direct information about individuals’ body shapes with AIS [35]. Since rib hump magnitude is not well correlated to radiological features, this would explain the poor correlation between cosmetic and radiological features, as indicated in our study.

So, for example, the Formetric Surface Topography system (DIERS Medical Systems, Inc. Chicago, IL, USA) could be used in future studies to create a 3D reconstruction of the spine and trunk shape instead of the X-ray’s 2D depiction [36]. Surface topography has been used as a radiation-free assessment tool to assess the posterior deformity and changes associated with scoliosis over time [36,37]. This would help to determine how strong the correlation is between body image disturbances and objective trunk shape analysis in AIS females.

D’Andrea et al. pointed out that deformities resulting from scoliosis, such as rib hump, trunk compensation, or waist asymmetry that affect appearance, become particularly important to patients in puberty [38]. Such deformities are an independent risk factor for developing self-image disturbances such as bulimia and anorexia nervosa [39]. According to studies designed by Fallstrom et al., only 8% of conservatively [40] treated patients and only 27% of surgically treated patients had a positive body image. Many studies have reported that dissatisfaction with appearance is more prevalent among adolescent girls than adolescent boys. Girls seem to be more interested in losing weight than boys, who have a greater willingness to increase muscle [40,41]. Risk analysis is critical in the determination of potential prevention. The role of mass media as a possible source of preferences for unrealistic attractiveness and predictor of body dissatisfaction is also widely documented [3]. Media influences have received a great deal of attention as the internalization of images and messages could be a causal risk factor [20]. Results from the current study indicate the potential importance of direct pressures generated by the media regarding appearance standards. In contrast to former studies, factors reflecting mass media as an essential source of information about attractiveness and internalization of an athletic ideal did not emerge in the current research [20,28]. The stronger endorsement of Internalization-General than Internalization-Athlete and Pressures was not consistent with the pattern shown among the American population [41]. However, our findings are consistent with Izydorczyk and Lizynczyk where the level of the Pressures and Internalization-General dominates over the intensity of other forms of sociocultural influences of the mass media on body image and physical appearance [32].

## 5. Conclusions

The presented study has shown that COPS and SATAQ-3 are tools with satisfactory statistical reliability. It confirmed the usefulness of the questionnaires in scientific research and adolescent screening. This procedure could indicate that patients who score high on COPS and SATAQ-3 are strongly determined to undergo AIS scoliosis treatment mostly due to aesthetical concerns. Moreover, further analyses and observations on the use of COPS and SATAQ-3 could also be conducted on various populations of Poles to enrich this tool with the development of norms for individual categories.

## Figures and Tables

**Figure 1 medicina-59-00581-f001:**
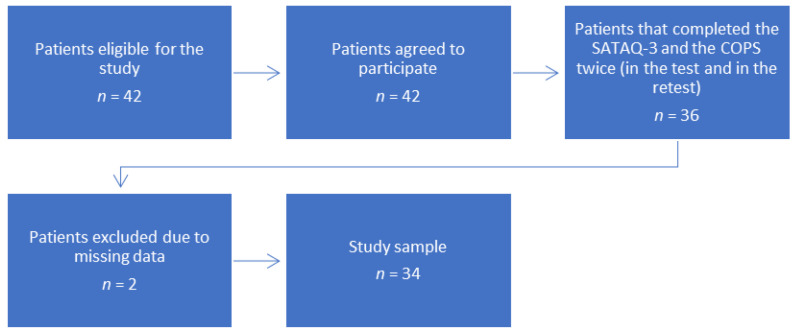
Patient flowchart.

**Table 1 medicina-59-00581-t001:** Characteristics of the study group.

Parameters	Mean (SD)	Range (Min–Max)	% (No.)
Age (years)	14.8 (1.8)	10–18	
Body Mass Index	18.59 (2.53)	13.62–28.08	
Cobb angle	26.9 (5.6)	20–38	
Apical translation of the central sacral vertical line (CSVL) (cm)	1.7 (0.9)	0.3–4.0	
Brace (in months)	24.9 (19.4)	3–72	
Brace (hours/day)	15.5 (3.5)	12–22	
Recreational sports activity (hours/week) *	5.2 (3.2)	1–13	
Physical education % (No.)			97.6 (33)
Recreational sports activity % (No.)			47.06 (22)
Place of residence			
Rural areas			50.00 (17)
City < 25,000			23.53 (8)
City > 25,000 < 200,000			17.65 (6)
City >200,000			8.82 (3)
School			
Primary			8.82 (3)
Secondary			58.82 (20)
Vocational			32.35 (11)

Note * out of participants practicing recreational sports activity.

**Table 2 medicina-59-00581-t002:** Distribution of results of COPS and SATAQ-3 in AIS patients.

	Test *				Retest **			
Parameters of COPS	Minimal Value	Maximum Value	Mean	SD	Minimal Value	Maximum Value	Mean	SD
Pie chart—1st Feature	0.15	1.00	0.56	0.25	0.25	1.00	0.58	0.27
Pie chart—2nd Feature	0.11	0.75	0.36	0.14	0.10	0.75	0.36	0.16
Pie chart—3rd Feature	0.05	0.50	0.20	0.13	0.05	0.35	0.19	0.10
Pie chart—4th Feature	0.12	0.35	0.23	0.10	0.15	0.30	0.24	0.06
Pie chart—5th Feature	0.20	0.20	0.20					
COPS total score	4.00	40.00	18.88	10.67	4.00	40.00	19.15	10.52
Parameters of SATAQ-3								
Internalization General	9.00	43.00	21.00	9.13	9.00	45.00	20.82	9.06
Internalization Athlete	4.00	25.00	13.00	5.05	5.00	25.00	13.15	5.31
Pressure	7.00	27.00	14.79	6.11	7.00	26.00	14.00	5.83
Information	9.00	38.00	23.53	7.06	9.00	45.00	23.38	8.74
Reverse keyed	8.00	37.00	21.41	6.53	9.00	40.00	22.09	6.70
SATAQ-3 total score	30.00	127.00	72.32	21.95	31.00	138.00	71.35	23.72

* the 1st completion of the COPS and SATAQ-3; ** the 2nd completion of the COPS and SATAQ-3.

**Table 3 medicina-59-00581-t003:** The percentage of concern allocated to each feature according to COPS in AIS patients.

Pie Chart—1st Feature	N	%
acne	1	2.94118
convex knees	1	2.94118
uneven hips	1	2.94118
pot belly	2	5.88235
scoliosis	12	35.29412
rib hump	5	14.70588
uneven waist	3	8.82353
uneven shoulders	3	8.82353
short legs	1	2.94118
nose	2	5.88235
uneven breast	1	2.94118
small lips	1	2.94118
big thigs	1	2.94118
Pie chart—2nd Feature		
large hips	5	14.70588
large birthmarks	1	2.94118
rib hump	1	2.94118
uneven shoulders	1	2.94118
round back	1	2.94118
uneven breasts	2	5.88235
uneven hips	2	5.88235
mammals	2	5.88235
obesity	1	2.94118
nose	1	2.94118
protruding head in relation to the body	1	2.94118
acne	1	2.94118
eyes with dark rings	1	2.94118
scoliosis	2	5.88235
big thigs	1	2.94118
vision defect	1	2.94118
crooked teeth	1	2.94118
too long legs	1	2.94118
uneven waist	2	5.88235
no-response	6	17.64706
Pie chart—3rd Feature		
pot belly	1	2.94118
scoliosis	1	2.94118
acne	3	8.82353
wide shoulders	1	2.94118
rib hump	1	2.94118
eyes with dark rings	1	2.94118
protruding ears	1	2.94118
uneven breasts	1	2.94118
crooked knees	1	2.94118
big buttocks	1	2.94118
uneven legs	2	5.88235
short stature	1	2.94118
big thigs	1	2.94118
uneven waist	2	5.88235
no-response	16	47.05882
Pie chart—4th Feature		
ugly teeth	2	5.88235
big nose	1	2.94118
acne	1	2.94118
no-response	30	88.23529
Pie chart—5th Feature		
big thigs	1	2.94118
no-response	33	97.05882

**Table 4 medicina-59-00581-t004:** Floor and ceiling effects of COPS and SATAQ-3 in AIS patients.

Test *			Retest **	
Parameters of COPS	*n*	%	*n*	%
Floor effect	3	8.82	2	5.89
Ceiling effect	2	5.89	1	2.94
Parameters of SATAQ-3				
Floor effect	1	2.94	1	2.94
Ceiling effect	1	2.94	1	2.94

* the 1st completion of the COPS and SATAQ-3; ** the 2nd completion of the COPS and SATAQ-3.

**Table 5 medicina-59-00581-t005:** The item–total correlation of the COPS in AIS patients.

	Test		Retest	
Items of COPS [21]	rS	*p* Value	rS	*p* Value
Item 1 * & TS	--	--	--	==
Item 2 & TS	rS = 0.19	*p* = 0.285	rS = 0.20	*p* = 0.268
Item 3 & TS	rS = 0.76	*p* < 0.0001	rS = 0.71	*p* < 0.00001
Item 4 & TS	rS = 0.72	*p* < 0.0001	rS = 0.86	*p* < 0.00001
Item 5 & TS	rS = 0.82	*p* < 0.00001	rS = 0.69	*p* < 0.00001
Item 6 & TS	rS = 0.71	*p* < 0.00001	rS = 0.78	*p* < 0.00001
Item 7 & TS	rS = 0.57	*p* < 0.001	rS = 0.50	*p* = 0.003
Item 8 & TS	rS = 0.46	*p* = 0.006	rS = 0.45	*p* = 0.007
Item 9 & TS	rS = 0.74	*p* < 0.0001	rS = 0.62	*p* < 0.0001
Item 10 & TS	rS = 0.68	*p* < 0.00001	rS = 0.68	*p* < 0.00001

Note TS—Total score; * item 1 refers to a verbal description of one’s feature(s) of a body which the person dislikes or would like to improve.

**Table 6 medicina-59-00581-t006:** The item–total correlation of the SATAQ-3 in AIS patients.

	Test		Retest	
Items of SATAQ-3 [20]	rS	*p* Value	rS	*p* Value
Item 1 & TS	rS = 0.49	*p* = 0.003	rS = 0.52	*p* = 0.002
Item 2 & TS	rS = 0.50	*p* = 0.003	rS = 0.64	*p* < 0.0001
Item 3 & TS	rS = 0.57	*p* < 0.001	rS = 0.39	*p* = 0.022
Item 4 & TS	rS = 0.78	*p* < 0.00001	rS = 0.82	*p* < 0.00001
Item 5 & TS	rS = 0.39	*p* = 0.021	rS = 0.75	*p* < 0.00001
Item 6 & TS	rS = 0.60	*p* < 0.0001	rS = 0.61	*p* < 0.0001
Item 7 & TS	rS = 0.57	*p* < 0.001	rS = 0.75	*p* < 0.00001
Item 8 & TS	rS = 0.70	*p* < 0.00001	rS = 0.68	*p* < 0.0001
Item 9 & TS	rS = 0.18	*p* = 0.300	rS = 0.52	*p* = 0.001
Item 10 & TS	rS = 0.59	*p* < 0.001	rS = 0.66	*p* < 0.0001
Item 11 & TS	rS = 0.74	*p* < 0.00001	rS = 0.80	*p* < 0.00001
Item 12 & TS	rS = 0.60	*p* < 0.001	rS = 0.53	*p* = 0.001
Item 13 &TS	rS = 0.38	*p* = 0.026	rS = 0.41	*p* = 0.017
Item 14 & TS	rS = 0.86	*p* < 0.00001	rS = 0.85	*p* < 0.00001
Item 15 & TS	rS = 0.63	*p* < 0.0001	rS = 0.72	*p* < 0.00001
Item 16 & TS	rS = 0.76	*p* < 0.00001	rS = 0.77	*p* < 0.00001
Item 17 & TS	rS = 0.66	*p* < 0.0001	rS = 0.71	*p* < 0.00001
Item 18 & TS	rS = 0.57	*p* < 0.0001	rS = 0.58	*p* < 0.001
Item 19 & TS	rS = 0.36	*p* = 0.039	rS = 0.31	*p* = 0.072
Item 20 & TS	rS = 0.67	*p* < 0.0001	rS = 0.57	*p* < 0.001
Item 21 & TS	rS = 0.63	*p* < 0.0001	rS = 0.64	*p* < 0.00001
Item 22 & TS	rS = 0.58	*p* < 0.0001	rS = 0.70	*p* < 0.00001
Item 23 & TS	rS = 0.46	*p* = 0.006	rS = 0.55	*p* < 0.001
Item 24 & TS	rS = 0.63	*p* < 0.0001	rS = 0.59	*p* < 0.001
Item 25 & TS	rS = 0.59	*p* < 0.001	rS = 0.72	*p* < 0.00001
Item 26 & TS	rS = 0.65	*p* < 0.0001	rS = 0.75	*p* < 0.00001
Item 27 & TS	rS = 0.70	*p* < 0.00001	rS = 0.58	*p* < 0.001
Item 28 & TS	rS = 0.32	*p* = 0.062	rS = 0.38	*p* = 0.026
Item 29 & TS	rS = 0.62	*p* < 0.0001	rS = 0.71	*p* < 0.00001
Item 30 & TS	rS = 0.36	*p* = 0.037	rS = 0.41	*p* = 0.015

TS—Total score.

**Table 7 medicina-59-00581-t007:** Correlation analysis by sociodemographic characteristics and COPS and SATAQ-3 in AIS patients.

	Age [Years]	Recreational Sports Activity (Hours/Week) *	Physical Education % (No.)	Place of Residence	School
COPS total score	rS = 0.05 *p* = 0.759	r = 0.18 *p* = 0.47	*p* = 0.903	*p* = 0.234	*p* = 0.545
SATAQ-3 total score	rS = 0.14 *p* = 0.425	r = 0.13 *p* = 0.607	*p* = 0.780	*p* = 0.119	*p* = 0.247

* out of participants practicing recreational sports activity.

**Table 8 medicina-59-00581-t008:** Correlation analysis by clinical and radiological parameters and COPS and SATAQ-3 in AIS patients.

	Body Mass Index	Cobb Angle	Apical Translation of the Central Sacral Vertical Line (CSVL) (cm)	Brace (in Months)	Brace (h/day)
COPS total score	rS = −0.23 *p* = 0.181	rS = 0.17 *p* = 0.267	r = 0.33 *p* = 0.06	rS = −0.16 *p* = 0.372	rS = 0.13 *p* = 0.474
SATAQ-3 total score	rS = −0.9 *p* = 0.624	rS = 0.01 *p* = 0.938	rS = 0.15 *p* = 0.8.55	rS = 0.06 *p* = 0.720	rS = −0.05 *p* = 0.796

## Data Availability

Data supporting aforementioned research are available from corresponding author.

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
