# Peer review of "Internalization of the Western Standard of Beauty and Body Satisfaction: Evaluation Utilizing COPS and SATAQ-3 Questionnaires among Girls with Scoliosis"

_medicina, 2023, doi:10.3390/medicina59030581_

Round 1
Reviewer 1 Report
Respected Authors,
Below are my comments and observations.
1. Title - Manuscript tiltle is too long.
2. Citations - Within the first 2 paragraphs, 17 references are cited! Cite which is relevant and recent.
3. Lines 60 - 67 should have been the whole introduction. Kindly explain it in detail with relevant references.
4. Line 109 - TH8. Mention it as 8th Thoracic vertebra or T8 (use subscript for 8).
5. Line 111 - 112 - What is the reason for the exclusion? Why was it not mentioned in the manuscript?
6. Line 124 - It is mentioned as 41 AIS subjects. But, 19+9+6=34. What about the remaining? Moreover, 41 is too low to represent the population of Polish.
7. Table 3 - Where is the Pie chart?
8. Line 279 - No direct proof available.
9. Line 310 - Totally contradictory to the title of this manuscript.
10 References - Several old references are mentioned. Use recent (2012 - 2023).
The aim or objective of this study is not clearly mentioned.
The title provides a broad idea that in this study, tools like COPS and SATAQ-3 are used to evaluate beauty and body satisfaction among girls suffering from AIS. But the contents in the conclusion states that COPS and SATAQ-3 are reliable statistical tools. This stuudy is not ablout the tools, but satisfaction among girls suffering from AIS.
The conclusions made in this manuscript does not have substantial evidence to support what they claim.
Regards.
Author Response
Authors' response to the review
Dear Editor,
We would like to thank you for giving us the opportunity to revise our manuscript. We have carefully read all the remarks made by the referees and addressed all of the reviewers’ comments.
In this cover letter we have given specific details on how we addressed each comment, directly after which we then cited the appropriate change made to the text. All the passages that we have changed or added to the manuscript, are marked up using the “Track Changes” function.
Once again, thank you for considering our manuscript.
REVIEWER’S COMMENTS Reviewer #1 and Reviewer #2:
Reviewer #1:
Comments and Suggestions for Authors
Review
- Title - Manuscript title is too long.
The title has been shortened as follows:
Internalization of the Western standard of beauty and body satisfaction: evaluation utilizing Cosmetic Procedure Screening Questionnaire and Sociocultural Attitudes Towards Appearance Questionnaire-3 among girls with scoliosis
- Citations - Within the first 2 paragraphs, 17 references are cited! Cite which is relevant and recent.
Citations have been updated to recent and relevant
- Lines 60 - 67 should have been the whole introduction. Kindly explain it in detail with relevant references.
We concur with the Reviewer’s opinion, that the Introduction is too long. According to this suggestion, and considering the other Reviewer’s specific suggestions referring to Introduction, we have, in general, shortened this part of manuscript to make it more concise for readers.
The corrected Introduction is now as follows:
- Introduction
Current research shows that sociocultural pressure to be thin and attractive could be the source of negative feelings about the body [1-4]. The internalization of media ideals and cultural pressure could quickly turn to body dissatisfaction concerns [5-7]. We must bear in mind, in the process of internalization the person adopts as his own certain behavior which previously appeared only as a direct reaction to environmental stimulation [8]. This adoption of body image created by media push women to body hyperawareness, as a result leading to body shame and social anxiety [9-11].
Among patients' motivations to undergo plastic surgery procedures thus, im-provement of their appearance, enhancement of self-confidence, self-esteem, and so-cial interactions are to most common. The vast majority of patients are satisfied with cosmetic surgery. However, a specific group of patients is not [12]. Dissatisfaction can occur due to a preexisting psychiatric condition disregarding clinical outcomes [13].
Adolescent idiopathic scoliosis is a common skeleton deformity with 2-3% preva-lence. A three-dimensional deformity occurs in: frontal (lateral bend), sagittal (deformation of thoracic kyphosis and lumbar lordosis), and transverse (associated with the rotation and translation of vertebrae) planes [14]. Progression of scoliosis is more prevalent in girls. As a result, girls need non-surgical or surgical treatment much more often than boys [15]. Progression of idiopathic scoliosis could manifest itself through multiple body deformities, including scapular and rib prominence, uneven shoulders, and an asymmetric waistline. According to several authors, those deformities can significantly influence self-esteem and body image [16-18]. Expressly, Auerbach et al. demonstrated more significant back-related body image disturbances in patients with scoliosis compared to healthy controls [12]. Spinal fusion surgery with instrumentation often reduces severe curves and minimizes the risk of curve progression [14-15]. Unfortunately, the technical success of the surgery does not correspond with patient satisfaction regarding surgical outcomes. Patient contentment with the surgical result appears to be unrelated to the medical aspects of spinal fusion surgery. This includes fundamental issues such as the preservation of pulmonary function and prevention of early osteoarthritis [19].
It should be emphasized that results concerning the body image assessment seem apparently contradictory. It was revealed that body dissatisfaction is related to the in-dividual's disfigured shape and weight. However, improving body shape and weight loss does not necessarily decrease body dissatisfaction [20]. This discrepancy might be explained by the Allocentric Lock Hypothesis (ALH) [21-23]. It implies that a basic disturbance may cause maintenance of body dissatisfaction in the way the body is re-membered [29]. Patients may be locked to negative remembrance of the body that is no more updated by contrasting self-centered representations driven by perception [24].
Returning to the main thread, it is still unclear under which circumstances pa-tients with AIS revalue the perception of scoliosis-related body deformation. It is also unclear how those patients experience body image after operative treatment and how this overestimation is characterized. Those findings suggest that patients with AIS might need support while changing their desired body shape to feel more optimistic about the cosmetic results of scoliosis treatment [23-24].
A preoperative assessment tool that helps to identify patients with underlying psychiatric issues, improper motivations, or unrealizable expectations may help avoid such situations, by providing a reason for psychological referral. However, it would not replace standard clinical examination but could alert the medical staff for possible psychological consultation.
The Sociocultural Attitudes Towards Appearance Questionnaire (SATAQ) measures awareness of the cultural ideal of beauty for women and the level of acceptance and internalization of that ideal. The recent version, SATAQ-3, is one of the gold instruments used worldwide [25]. The SATAQ-3 has shown a relatively stable internal structure and boasts good indicators of reliability and validity when applied to Western women [25].
Another helpful questionnaire proven to be a valid, feasible, and reliable screening tool for patients is the Cosmetic Procedure Screening Questionnaire (COPS). It can be applied to cosmetic practice before performing any intervention. The scale has a high sensitivity for diagnosing Body Dysmorphic Disorder (BDD) in people likely to undergo a cosmetic procedurę [26].
Therefore, our study aimed to adapt and analyze psychometric properties of the Polish ver-sions of COPS and SATAQ-3, which could identify AIS patients especially threatened with body image disorders and might predict dissatisfaction with a desired by patients cosmetic result of brace treatment. The study represents a preliminary validation.
- Line 109 - TH8. Mention it as 8th Thoracic vertebra or T8 (use subscript for 8).
It has been modified into T8
- Line 111 - 112 - What is the reason for the exclusion? Why was it not mentioned in the manuscript?
Sufficient explanation has been introduced to the text.
According to Sponseller, patients with a high thoracic curve and a major curve apex above Th8 were excluded [27]. Those patients could not be treated with thoraco-lumbo-sarcal orthoses used in current study. The exclusion criteria were also mental disorders associated with developing a distorted body image (psychotic, dysmorphia). None of the patients had received any other spinal treatment before.
- Line 124 - It is mentioned as 41 AIS subjects. But, 19+9+6=34. What about the remaining? Moreover, 41 is too low to represent the population of Polish.
The erroneous number of subjects have been correted to 34.
We agree that study sample of 34 patients is not enough to represent the population of Polish.
Thus, we have corrected our study aims, as follows:
Therefore, our study aimed to adapt and analyze psychometric properties of the Polish ver-sions of COPS and SATAQ-3, which could identify AIS patients especially threatened with body image disorders and might predict dissatisfaction with a desired by patients cosmetic result of brace treatment. The study represents a preliminary validation.
- Table 3 - Where is the Pie chart?
The Pie chart is the part of COPS questionnaire, which is in our opinion described sufficiently in the text.
Table 3 presents the percentage of concern allocated to each Feature according to COPS (from the pie chart regarding the 1st Feature to 5th Feature).
- Line 279 - No direct proof available.
Our results are as follows:
However, SATAQ-3 total score of over 60 points, indicating a high level of Internalization, applied to 67,64% of patients in the test (23 participants) and 61,76% of patients in the retest (21 participants).
- Line 310 - Totally contradictory to the title of this manuscript.
We would like to explain that the presented research includes cultural adaptation of the questionnaires thus the conclusion about the reliability corresponds to the aim of the study.
We adopted the SATAQ-3 and COPS to Polish cultural settings. The process of cultural adaptation of the questionnaires was compliant with the guidelines of the International Quality of Life Assessment (IQOLA) Project [37].
10 References - Several old references are mentioned. Use recent (2012 - 2023).
References were updated to more recent.
11 The aim or objective of this study is not clearly mentioned
The aim is mentioned in the last paragraph of the introduction as follows:
Our study aimed to adapt and analyze psychometric properties of the Polish versions of COPS and SATAQ-3, which could identify AIS patients especially threatened with body image disorders and might predict dissatisfaction with a desired by patients cosmetic result of brace treatment. The study represents a preliminary validation.
12 The title provides a broad idea that in this study, tools like COPS and SATAQ-3 are used to evaluate beauty and body satisfaction among girls suffering from AIS. But the contents in the conclusion states that COPS and SATAQ-3 are reliable statistical tools. This study is not about the tools, but satisfaction among girls suffering from AIS.
We would like to explain that the presented research includes cultural adaptation of the questionnaires thus the conclusion about the reliability corresponds to the aim of the study.
13 The conclusions made in this manuscript does not have substantial evidence to support what they claim.
The conclusions have been rewrited as follows.
Conclusions
The presented study has shown that COPS and SATAQ-3 are tools with satisfactory reliability. It confirmed the usefulness of those questionnaires in scientific research and adolescent screening. This procedure could indicate patients scoring high on COPS and SATAQ-3, who are strongly determined to undergo AIS scoliosis treatment mostly due to aesthetical concerns. Moreover, further analyses and observations on use of COPS and SATAQ-3 could also be conducted on various, clinical and general, populations of Poles to enrich this tool with the development of norms for individual categories.
Reviewer #2:
Comments and Suggestions for Authors
Review
- I would better describe what is “internalization” in the intro.
We would like to thank the Reviewer for his thoughtful analysis of our manuscript. We concur with the Reviewer’s opinion, that the term internalization needs a better description in the Introduction. We have supplemented the Introduction section, as follows:
Introduction
Current research shows that sociocultural pressure to be thin and attractive could be the source of negative feelings about the body [1-4]. The internalization of media ideals and cultural pressure could quickly turn to body dissatisfaction concerns [5-7]. We must bear in mind, in the process of internalization the person adopts as his own certain behavior which previously appeared only as a direct reaction to environmental stimulation [8]. This adoption of body image created by media push women to body hyperawareness, as a result leading to body shame and social anxiety [9-11].
- Please shorten the SATAQ and COPS description in the intro. Part of it could be transferred to the discussion.
According to this suggestion, we have shortened the Introduction to make it more concise for readers. We have also supplemented the Discussion section with the SATAQ and COPS description.
The corrections are as follows:
Introduction (…)
The Sociocultural Attitudes Towards Appearance Questionnaire (SATAQ) measures awareness of the cultural ideal of beauty for women and the level of acceptance and internalization of that ideal. The recent version, SATAQ-3, is one of the gold instruments used worldwide [25]. The SATAQ-3 has shown a relatively stable internal structure and boasts good indicators of reliability and validity when applied to Western women [25].
Another helpful questionnaire proven to be a valid, feasible, and reliable screening tool for patients is the Cosmetic Procedure Screening Questionnaire (COPS). It can be applied to cosmetic practice before performing any intervention. The scale has a high sensitivity for diagnosing Body Dysmorphic Disorder (BDD) in people likely to undergo a cosmetic procedurę [26].
Discussion
Research has demonstrated that body image is a significant concern among girls and women [33]. Many female adolescents attempt to have a perfect body appearance. Unfortunately, if they do not achieve their ideal appearance, they could develop negative emotions, leading to stress and anxiety [5]. Also mass media play a crucial role in distributing these body image criteria [5-34]. Thus, the psychometric assessment of body image could play a significant role in the preliminary selection of those who should be referred to cognitive behavior therapy [35-36].
Referring to those assessment tools, Thompson et al. recommended the utilization of SATAQ-3 because of its excellent psychometric properties, the availability of its translated versions in different languages, and its structure dimensions [25]. However, many researchers use this scale mainly to study the relationship between sociocultural appearance standards promoted in the media and selected risk factors concerning eating disorders [37]. The first Polish attempts to use SATAQ-3 took place in the research of Izydorczyk and Lizynczyk [38]. So far, there has been no Polish standardization of a tool for measuring body image and physical appearance standards promoted by mass media.
Also COPS, recognized as reliable screening tool, is sensitive to changes in patients under cognitive-behavior therapy. It may be used as an outcome measure after treatment to assess if there is any improvement in symptoms of BDD. [38].
However, it was found in our studies that in general, AIS patients perceived their appearance positively. These findings are similar to results presented by Misterska et al. where as many as 60% of the study sample would not change anything [16]. Interestingly a COPS total score ≥ 40 points, indicating a higher level of BDD applied to 5,88% of participants and 2,94% in the retest, which is above general population estimation, but still significantly lower than in cosmetic surgery patients. [35-36]. As previously stated, until recently, there were few studies that examined the perception of deformities in scoliosis patients [39-40]. This is probably because it is difficult to objectively assess the self-image perceptions of scoliotic patients.
- L101 remove « by patients”
The correction is as follows:
Our study aimed to adapt and analyze psychometric properties of the Polish versions of COPS and SATAQ-3, which could identify AIS patients especially threatened with body image disorders and might predict dissatisfaction with a desired cosmetic result of brace treatment.
- M&M. Since you refer in the results to different items of each questionnaire, I feel it is important to place the Questionnaires in M&M.
We would like to thank the Reviewer for his careful review. We agree with Reviewer’s opinion that Tables in Results section are mostly unreadable since it is not possible to understand each item signification. Therefore, we are of opinion that supplementing Table 5 and Table 6 with the content of particular items of the COPS and SATAQ-3 would facilitate the readers a better insight of the study Results.
The corrected Table 5 and Table 6 are now as follows:
Table 5. The item-total correlation of the COPS in AIS patients.
|
|
Test |
Retest |
||
|
Items of COPS [35] |
rS |
p value |
rS |
p value |
|
Item 1 * & TS |
-- |
-- |
-- |
== |
|
Item 2 How often do you deliberately check your feature(s)? Not accidentally catch sight of it.& TS |
rS=0,19 |
p=0,285 |
rS=0,20 |
p=0,268 |
|
Item 3 How much do you feel your feature(s) is currently ugly, unattractive or ‘not right’?& TS |
rS=0,76 |
p<0,0001 |
rS=0,71 |
p<0,00001 |
|
Item 4 How much does your feature(s) currently cause you a lot of distress? & TS |
rS=0,72 |
p<0,0001 |
rS=0,86 |
p<0,00001 |
|
Item 5 How often does your feature(s) currently lead you to avoid situations or activities?& TS |
rS=0,82 |
p<0,00001 |
rS=0,69 |
p<0,00001 |
|
Item 6 How much does your feature(s) currently preoccupy you? That is, you think about it a lot and it is hard to stop thinking about it? & TS |
rS=0,71 |
p<0,00001 |
rS=0,78 |
p<0,00001 |
|
Item 7 If you have a partner, how much does your feature(s) currently have an effect on your relationship with an existing partner? If you do not have a partner, how much does it have an effect on dating or developing a relationship?& TS |
rS=0,57 |
p<0,001 |
rS=0,50 |
p=0,003 |
|
Item 8 How much does your feature(s) currently interfere with your ability to work or study, or your role as a homemaker? (Please rate this even if you are not working or studying: we are interested in your ability to work or study.)& TS |
rS=0,46 |
p=0,006 |
rS=0,45 |
p=0,007 |
|
Item 9 How much does your feature(s) currently interfere with your social life?& TS |
rS=0,74 |
p<0,0001 |
rS=0,62 |
p<0,0001 |
|
Item 10 How much do you feel your appearance is the most important aspect of who you are?& TS |
rS=0,68 |
p<0,00001 |
rS=0,68 |
p<0,00001 |
Note TS – total score; * item 1 refers to a verbal description of one’s feature(s) of a body which the person dislikes or would like to improve.
Table 6. The item-total correlation of the SATAQ in AIS patients.
|
|
Test |
Retest |
||
|
Items of SATAQ [30] |
rS |
p value |
rS |
p value |
|
Item 1 TV programs are an important source of information about fashion and "being attractive & TS |
rS=0,49 |
p=0,003 |
rS=0,52 |
p=0,002 |
|
Item 2 I've felt pressure from TV or magazines to lose weight.& TS |
rS=0,50 |
p=0,003 |
rS=0,64 |
p<0,0001 |
|
Item 3 I do not care if my body looks like the body of people who are on TV.& TS |
rS=0,57 |
p<0,001 |
rS=0,39 |
p=0,022 |
|
Item 4 I compare my body to the bodies of people who are on TV.& TS |
rS=0,78 |
p<0,00001 |
rS=0,82 |
p<0,00001 |
|
Item 5 TV commercials are an important source of information about fashion and "being attractive & TS |
rS=0,39 |
p=0,021 |
rS=0,75 |
p<0,00001 |
|
Item 6 I do not feel pressure from TV or magazines to look pretty.& TS |
rS=0,60 |
p<0,0001 |
rS=0,61 |
p<00001 |
|
Item 7 I would like my body to look like the models who appear in magazines.& TS |
rS=0,57 |
p<0,001 |
rS=0,75 |
p<0,00001 |
|
Item 8 I compare my appearance to the appearance of TV and movie stars.& TS |
rS=0,70 |
p<0,00001 |
rS=0,68 |
p<0,0001 |
|
Item 9 Music videos on TV are not an important source of information about fashion and "being attractive."& TS |
rS=0,18 |
p=0,300 |
rS=0,52 |
p=0,001 |
|
Item 10 I've felt pressure from TV and magazines to be thin.& TS |
rS=0,59 |
p<0,001 |
rS=0,66 |
p<0,0001 |
|
Item 11 I would like my body to look like the people who are in movies.& TS |
rS=0,74 |
p<0,00001 |
rS=0,80 |
p<0,00001 |
|
Item 12 I do not compare my body to the bodies of people who appear in magazines.& TS |
rS=0,60 |
p<0,001 |
rS=0,53 |
p=0,001 |
|
Item 13 Magazine articles are not an important source of information about fashion and "being attractive." &TS |
rS=0,38 |
p=0,026 |
rS=0,41 |
p=0,017 |
|
Item 14 I've felt pressure from TV or magazines to have a perfect body & TS |
rS=0,86 |
p<0,00001 |
rS=0,85 |
p<0,00001 |
|
Item 15 I wish I looked like the models in music videos & TS |
rS=0,63 |
p<0,0001 |
rS=0,72 |
p<0,00001 |
|
Item 16 I compare my appearance to the appearance of people in magazines & TS |
rS=0,76 |
p<0,00001 |
rS=0,77 |
p<0,00001 |
|
Item 17 Magazine advertisements are an important source of information about fashion and "being attractive" & TS |
rS=0,66 |
p<0,0001 |
rS=0,71 |
p<0,00001 |
|
Item 18 I've felt pressure from TV or magazines to diet & TS |
rS=0,57 |
p<0,0001 |
rS=0,58 |
p<0,001 |
|
Item 19 I do not wish to look as athletic as the people in magazines & TS |
rS=0,36 |
p=0,039 |
rS=0,31 |
p=0,072 |
|
Item 20 I compare my body to that of people in "good shape” & TS |
rS=0,67 |
p<0,0001 |
rS=0,57 |
p<0,001 |
|
Item 21 Pictures in magazines are an important source of information about fashion and "being attractive" & TS |
rS=0,63 |
p<0,0001 |
rS=0,64 |
p<0,00001 |
|
Item 22 I've felt pressure from TV or magazines to exercise & TS |
rS=0,58 |
p<0,0001 |
rS=0,70 |
p<0,00001 |
|
Item 23 I wish I looked as athletic as sports stars & TS |
rS=0,46 |
p=0,006 |
rS=0,55 |
p<0,001 |
|
Item 24 I compare my body to that of people who are athletic & TS |
rS=0,63 |
p<0,0001 |
rS=0,59 |
p<0,001 |
|
Item 25 Movies are an important source of information about fashion and "being attractive"& TS |
rS=0,59 |
p<0,001 |
rS=0,72 |
p<0,00001 |
|
Item 26 I've felt pressure from TV or magazines to change my appearance & TS |
rS=0,65 |
p<0,0001 |
rS=0,75 |
p<0,00001 |
|
Item 27 I do not try to look like the people on TV & TS |
rS=0,70 |
p<0,00001 |
rS=0,58 |
p<0,001 |
|
Item 28 Movie starts are not an important source of information about fashion and "being attractive” & TS |
rS=0,32 |
p=0,062 |
rS=0,38 |
p=0,026 |
|
Item 29 Famous people are an important source of information about fashion and "being attractive"& TS |
rS=0,62 |
p<0,0001 |
rS=0,71 |
p<0,00001 |
|
Item 30 I try to look like sports athletes & TS |
rS=0,36 |
p=0,037 |
rS=0,41 |
p=0,015 |
Note TS – total score;
- Please clarify the method with sub-sections. It is really difficult to understand the results without a better description of what was measured and computed.
We understand that the Reviewer would like us to reorganize the Methods section to make it more clear for readers. The corrections are as follows:
2.3. Cultural adaptation of original versions of COPS and SATAQ-3
We adopted the SATAQ-3 and COPS to Polish cultural settings. The process of cultural adaptation of the questionnaires was compliant with the guidelines of the International Quality of Life Assessment (IQOLA) Project [37] and comprises of translation procedures and analysis of psychometric properties of adapted questionnaires.
2.3.1 Questionnaires
The COPS screens for attribute (s) that the surveyed finds unattractive, the characteristics of the cosmetic procedures and diagnostic criteria of BDD. Answers to questions are made by a Likert scale which helps the surveyed to quantify responses. The total score is calculated by summing questions from 2 to 10 (items 2, 3 and 5 are reversed). It ranges from 0 to 72 with a cut-off value of ≥ 40, indicating a high probability of having BDD [29]. Those individuals should be referred for further assessment. The scale is available online to download for free [30]. The aforementioned questionnaire is sensitive to changes and has good test-retest reliability, convergent validity and strong internal consistency [29].
The SATAQ-3 is a 30-item scale with four theoretical subscales. The first subscale, with nine items, is Internalization-General and evaluates general media impact related to TV, newspapers, and movies. The second subscale, with five items, is Internalization-Athlete and evaluates the Internalization of gymnastic and sports models. The other subscales are Information, with nine items. It evaluates how far it is recognized that different media are considered significant sources of information about look. The subscale Pressures with seven items, evaluates indvidual feelings of pressure from display to media images and messages to modify one's look [25]. The question format is a Likert-type scale ranging from 1 = completely disagree to 5 = completely agree. The total ranges from 30 to 150 with a cut-off value of ≥ 60, indicating a high probability of having BDD [25].
2.3.2. Translation procedure
The expert committee cooperated with the researchers to translate COPS and SATAQ-3, using the forward-backward method. At first, two experts working separately, translated the questionnaire into Polish. Polish was the native language of these translators. In the second stage, these translations were synthesized into one version by translators and authors of the project. Then, two native English speakers who were bilingual back-translated COPS and SATAQ-3 into English. These translators did not know the original English version. Finally, all translators, a clinical psychologist, a statistician and orthopedic surgeons reviewed all the translations. A consensus concerning all the inconsistencies has been reached, establishing the Polish version of the SATAQ-3 and COPS. Finally, study participants filled in Polish versions of COPS and SATAQ-3 twice.
2.3.3. Statistics
To determine the psychometric properties of Polish versions of COPS and SATAQ-3, statistical analysis was performed. Descriptive statistics (mean, 95 % confidence intervals, range and standard deviations) were utilized to describe the distribution of the results with respect to quantitative statistical features. With respect to qualitative features, we assigned percentages to the number of units belonging to the described categories of a given feature.
In addition, the psychometric properties of the COPS and SATAQ-3 were determined by the distribution of scores and determination of the floor effect (% of patients with the minimum score) and ceiling effect (% of patients with the maximum score).
Internal consistency was examined using Cronbach's alpha coefficient. Cronbach's alpha coefficient values were accepted as follows: excellent > 0.80, adequate 0.70–0.79 and poor < 0.70 [32]. To assess temporal stability, test-retest method was used. The COPS and SATAQ-3, as pointed above, were filled in twice and then the Intraclass Correlation Coefficient (ICC) was calculated. Discriminant validity was assessed by calculating the correlation between the total COPS and SATAQ-3, and the Cobb angle.
In addition, the responsiveness of the COPS and SATAQ-3 were determined by means of effect sizes, which were calculated for each measure by dividing the mean absolute change score by the standard deviation of the baseline score.The interpretation of the magnitude of the effect size was based on Cohen's rule-of-thumb, in which an effect size of 0.2–0.5 was considered as small, 0.5–0.8 as moderate and over 0.8 or greater represented a large effect.
- Please explain “retest” in the tables. Have sent the questionnaire twice ? If so please revise M&M.
We would like to thank the Reviewer for his careful review. Yes, the questionnaires were filled in twice, to assess one of the psychometric properties, test-restesr reliability of Polish versions od COPS and SATAQ-3. We have corrected Material and Methods section and clarified this issue to make it more clear for readers, as follows:
Study design
Concerning the bracing protocol, all patients have been initially prescribed full-time bracing. The data regarding brace-wearing time compliance was based on anamnesis with patients during the clinical examination every three months. To verify patient claims concerning the time of bracing, an additional interview was conducted with the parents. Regarding the study protocol, all the participants and their parents received thorough information on the aim of the study. They were assured of anonymity, following which they gave informed consent. Then they were asked to fill out the study questionnaires twice at home, with a 2-days interval, and return them by mail.
Statistics
Internal consistency was examined using Cronbach's alpha coefficient. Cronbach's alpha coefficient values were accepted as follows: excellent > 0.80, ade-quate 0.70–0.79 and poor < 0.70 [32]. To assess temporal stability, test-retest method was used. The COPS and SATAQ-3 were filled in twice and then the Intraclass Correlation Coefficient (ICC) was calculated.
Table 2. Distribution of results of COPS and SATAQ-3 in AIS patients.
|
|
Test * |
Retest** |
||||||
|
Parameters of COPS |
Minimal value |
Maximum value |
Mean |
SD |
Minimal value |
Maximum value |
Mean |
SD |
|
Pie chart- 1st Feature |
0,15 |
1,00 |
0,56 |
0,25 |
0,25 |
1,00 |
0,58 |
0,27 |
|
Pie chart- 2nd Feature |
0,11 |
0,75 |
0,36 |
0,14 |
0,10 |
0,75 |
0,36 |
0,16 |
|
Pie chart- 3rd Feature |
0,05 |
0,50 |
0,20 |
0,13 |
0,05 |
0,35 |
0,19 |
0,10 |
|
Pie chart- 4th Feature |
0,12 |
0,35 |
0,23 |
0,10 |
0,15 |
0,30 |
0,24 |
0,06 |
|
Pie chart- 5th Feature |
0,20 |
0,20 |
0,20 |
|
|
|
|
|
|
COPS total score |
4,00 |
40,00 |
18,88 |
10,67 |
4,00 |
40,00 |
19,15 |
10,52 |
|
Parameters of SATAQ-3 |
|
|
|
|
|
|
|
|
|
Internalization General |
9,00 |
43,00 |
21,00 |
9,13 |
9,00 |
45,00 |
20,82 |
9,06 |
|
Internalization Athlete |
4,00 |
25,00 |
13,00 |
5,05 |
5,00 |
25,00 |
13,15 |
5,31 |
|
Pressure |
7,00 |
27,00 |
14,79 |
6,11 |
7,00 |
26,00 |
14,00 |
5,83 |
|
Information |
9,00 |
38,00 |
23,53 |
7,06 |
9,00 |
45,00 |
23,38 |
8,74 |
|
Reverse keyed |
8,00 |
37,00 |
21,41 |
6,53 |
9,00 |
40,00 |
22,09 |
6,70 |
|
SATAQ-3 total score |
30,00 |
127,00 |
72,32 |
21,95 |
31,00 |
138,00 |
71,35 |
23,72 |
Note * the 1st completion of the COPS and SATAQ-3; ** the 2nd completion of the COPS and SATAQ-3
Table 4. Floor and ceiling effects of COPS and SATAQ-3 in AIS patients.
|
Test* Retest** |
||||
|
Parameters of COPS |
n |
% |
n |
% |
|
Floor effect |
3 |
8.82 |
2 |
5.89 |
|
Ceiling effect |
2 |
5,89 |
1 |
2.94 |
|
Parameters of SATAQ-3 |
|
|
|
|
|
Floor effect |
1 |
2.94 |
1 |
2.94 |
|
Ceiling effect |
1 |
2.94 |
1 |
2.94 |
Note * the 1st completion of the COPS and SATAQ-3; ** the 2nd completion of the COPS and SATAQ-3
Table 5. The item-total correlation of the COPS in AIS patients.
|
|
Test* |
Retest** |
||
|
Items of COPS [35] & TS *** |
rS |
p value |
rS |
p value |
|
Item 1 & TS**** |
-- |
-- |
-- |
== |
|
Item 2 How often do you deliberately check your feature(s)? Not accidentally catch sight of it.& TS |
rS=0,19 |
p=0,285 |
rS=0,20 |
p=0,268 |
|
Item 3 How much do you feel your feature(s) is currently ugly, unattractive or ‘not right’?& TS |
rS=0,76 |
p<0,0001 |
rS=0,71 |
p<0,00001 |
|
Item 4 How much does your feature(s) currently cause you a lot of distress? & TS |
rS=0,72 |
p<0,0001 |
rS=0,86 |
p<0,00001 |
|
Item 5 How often does your feature(s) currently lead you to avoid situations or activities?& TS |
rS=0,82 |
p<0,00001 |
rS=0,69 |
p<0,00001 |
|
Item 6 How much does your feature(s) currently preoccupy you? That is, you think about it a lot and it is hard to stop thinking about it? & TS |
rS=0,71 |
p<0,00001 |
rS=0,78 |
p<0,00001 |
|
Item 7 If you have a partner, how much does your feature(s) currently have an effect on your relationship with an existing partner? If you do not have a partner, how much does it have an effect on dating or developing a relationship?& TS |
rS=0,57 |
p<0,001 |
rS=0,50 |
p=0,003 |
|
Item 8 How much does your feature(s) currently interfere with your ability to work or study, or your role as a homemaker? (Please rate this even if you are not working or studying: we are interested in your ability to work or study.)& TS |
rS=0,46 |
p=0,006 |
rS=0,45 |
p=0,007 |
|
Item 9 How much does your feature(s) currently interfere with your social life?& TS |
rS=0,74 |
p<0,0001 |
rS=0,62 |
p<0,0001 |
|
Item 10 How much do you feel your appearance is the most important aspect of who you are?& TS |
rS=0,68 |
p<0,00001 |
rS=0,68 |
p<0,00001 |
Note * the 1st completion of the COPS and SATAQ-3; ** the 2nd completion of the COPS and SATAQ-3; *** item 1 refers to a verbal description of one’s feature(s) of a body which the person dislikes or would like to improve; TS- total score
Table 6. The item-total correlation of the SATAQ in AIS patients.
|
|
Test* |
Retest** |
||
|
Items of SATAQ [30]&TS*** |
rS |
p value |
rS |
p value |
|
Item 1 TV programs are an important source of information about fashion and "being attractive & TS |
rS=0,49 |
p=0,003 |
rS=0,52 |
p=0,002 |
|
Item 2 I've felt pressure from TV or magazines to lose weight.& TS |
rS=0,50 |
p=0,003 |
rS=0,64 |
p<0,0001 |
|
Item 3 I do not care if my body looks like the body of people who are on TV.& TS |
rS=0,57 |
p<0,001 |
rS=0,39 |
p=0,022 |
|
Item 4 I compare my body to the bodies of people who are on TV.& TS |
rS=0,78 |
p<0,00001 |
rS=0,82 |
p<0,00001 |
|
Item 5 TV commercials are an important source of information about fashion and "being attractive & TS |
rS=0,39 |
p=0,021 |
rS=0,75 |
p<0,00001 |
|
Item 6 I do not feel pressure from TV or magazines to look pretty.& TS |
rS=0,60 |
p<0,0001 |
rS=0,61 |
p<00001 |
|
Item 7 I would like my body to look like the models who appear in magazines.& TS |
rS=0,57 |
p<0,001 |
rS=0,75 |
p<0,00001 |
|
Item 8 I compare my appearance to the appearance of TV and movie stars.& TS |
rS=0,70 |
p<0,00001 |
rS=0,68 |
p<0,0001 |
|
Item 9 Music videos on TV are not an important source of information about fashion and "being attractive."& TS |
rS=0,18 |
p=0,300 |
rS=0,52 |
p=0,001 |
|
Item 10 I've felt pressure from TV and magazines to be thin.& TS |
rS=0,59 |
p<0,001 |
rS=0,66 |
p<0,0001 |
|
Item 11 I would like my body to look like the people who are in movies.& TS |
rS=0,74 |
p<0,00001 |
rS=0,80 |
p<0,00001 |
|
Item 12 I do not compare my body to the bodies of people who appear in magazines.& TS |
rS=0,60 |
p<0,001 |
rS=0,53 |
p=0,001 |
|
Item 13 Magazine articles are not an important source of information about fashion and "being attractive." &TS |
rS=0,38 |
p=0,026 |
rS=0,41 |
p=0,017 |
|
Item 14 I've felt pressure from TV or magazines to have a perfect body & TS |
rS=0,86 |
p<0,00001 |
rS=0,85 |
p<0,00001 |
|
Item 15 I wish I looked like the models in music videos & TS |
rS=0,63 |
p<0,0001 |
rS=0,72 |
p<0,00001 |
|
Item 16 I compare my appearance to the appearance of people in magazines & TS |
rS=0,76 |
p<0,00001 |
rS=0,77 |
p<0,00001 |
|
Item 17 Magazine advertisements are an important source of information about fashion and "being attractive" & TS |
rS=0,66 |
p<0,0001 |
rS=0,71 |
p<0,00001 |
|
Item 18 I've felt pressure from TV or magazines to diet & TS |
rS=0,57 |
p<0,0001 |
rS=0,58 |
p<0,001 |
|
Item 19 I do not wish to look as athletic as the people in magazines & TS |
rS=0,36 |
p=0,039 |
rS=0,31 |
p=0,072 |
|
Item 20 I compare my body to that of people in "good shape” & TS |
rS=0,67 |
p<0,0001 |
rS=0,57 |
p<0,001 |
|
Item 21 Pictures in magazines are an important source of information about fashion and "being attractive" & TS |
rS=0,63 |
p<0,0001 |
rS=0,64 |
p<0,00001 |
|
Item 22 I've felt pressure from TV or magazines to exercise & TS |
rS=0,58 |
p<0,0001 |
rS=0,70 |
p<0,00001 |
|
Item 23 I wish I looked as athletic as sports stars & TS |
rS=0,46 |
p=0,006 |
rS=0,55 |
p<0,001 |
|
Item 24 I compare my body to that of people who are athletic & TS |
rS=0,63 |
p<0,0001 |
rS=0,59 |
p<0,001 |
|
Item 25 Movies are an important source of information about fashion and "being attractive"& TS |
rS=0,59 |
p<0,001 |
rS=0,72 |
p<0,00001 |
|
Item 26 I've felt pressure from TV or magazines to change my appearance & TS |
rS=0,65 |
p<0,0001 |
rS=0,75 |
p<0,00001 |
|
Item 27 I do not try to look like the people on TV & TS |
rS=0,70 |
p<0,00001 |
rS=0,58 |
p<0,001 |
|
Item 28 Movie starts are not an important source of information about fashion and "being attractive” & TS |
rS=0,32 |
p=0,062 |
rS=0,38 |
p=0,026 |
|
Item 29 Famous people are an important source of information about fashion and "being attractive"& TS |
rS=0,62 |
p<0,0001 |
rS=0,71 |
p<0,00001 |
|
Item 30 I try to look like sports athletes & TS |
rS=0,36 |
p=0,037 |
rS=0,41 |
p=0,015 |
Note * the 1st completion of the COPS and SATAQ-3; ** the 2nd completion of the COPS and SATAQ-3; *** item 1 refers to a verbal description of one’s feature(s) of a body which the person dislikes or would like to improve; TS- total score
- Tables are mostly unreadable since it is not possible to understand each item signification.
We agree with the Reviewer’s opinion. As pointed above, we have supplemented Table 5 and Table 6 with the individual item meanings.
- The first 2 paragraphs repeat the intro. I recommend focusing on the study results.
We agree with this opinion. According to Reviewer’s advice, we have shortened the Discussion section to focus directly on our study Results.
Discussion
Research has demonstrated that body image is a significant concern among girls and women [33]. Many female adolescents attempt to have a perfect body appearance. Unfortunately, if they do not achieve their ideal appearance, they could develop negative emotions, leading to stress and anxiety [5]. Also mass media play a crucial role in distributing these body image criteria [5,34]. Thus, the psychometric assessment of body image could play a significant role in the preliminary selection of those who should be referred to cognitive behavior therapy [35-36].
Referring to those assessment tools, Thompson et al. recommended the utilization of SATAQ-3 because of its excellent psychometric properties, the availability of its translated versions in different languages, and its structure dimensions [25]. However, many researchers use this scale mainly to study the relationship between sociocultural appearance standards promoted in the media and selected risk factors concerning eating disorders [37]. The first Polish attempts to use SATAQ-3 took place in the research of Izydorczyk and Lizynczyk [38]. So far, there has been no Polish standardization of a tool for measuring body image and physical appearance standards promoted by mass media.
Also COPS, recognized as reliable screening tool, is sensitive to changes in patients under cognitive-behavior therapy. It may be used as an outcome measure after treatment to assess if there is any improvement in symptoms of BDD. [38].
However, it was found in our studies that in general, AIS patients perceived their appearance positively. These findings are similar to results presented by Misterska et al. where as many as 60% of the study sample would not change anything [16]. Interestingly a COPS total score ≥ 40 points, indicating a higher level of BDD applied to 5,88% of participants and 2,94% in the retest, which is above general population estimation, but still significantly lower than in cosmetic surgery patients. [35-36]. As previously stated, until recently, there were few studies that examined the perception of deformities in scoliosis patients [39-40]. This is probably because it is difficult to objectively assess the self-image perceptions of scoliotic patients.
- One of the main limitations of the study is that there was no objective trunk shape analysis. Rib hump magnitude for instance is not well correlated to radiological features, which may explain the poor correlation between cosmetic and radiological features. 3D surface topography analysis should be discussed. There is a high correlation between self-image and trunk shape.
We agree with the Reviewer’s opinion that, referring to our study limitations, we did not discuss the lack of objective trunk shape analysis. Considering this issue, we have supplemented the Discussion section, as follows:
Discussion
The present research proves that the sociocultural appearance standards, promoted in mass media play an essential role in the studied group. In particular, the findings revealed that internalization of the Western standard of beauty and body satisfaction among adolescent girls with idiopathic scoliosis is neither correlated in regards to sociodemographics nor to clinical parameters. Moreover, our study revealed that radiological parameters, which play a crucial role in outcome measurement and potential further surgical treatment are not correlated with COPS and SATAQ-3 scores. However, body image in scoliosis patients significantly affects appearance perception [42].
On the other hand, accounting to our study limitations, we did not perform an objective trunk shape analysis. Only a standard, radiographic assessment was utilized. Meanwhile, we must bear in mind that radiographs do not provide direct information about individuals' body shapes with AIS [43,44]. Since rib hump magnitude is not well correlated to radiological features, this would explain the poor correlation between cosmetic and radiological features, as indicated in our study.
So, for example, the Formetric Surface Topography system (DIERS Medical Systems, Inc. Chicago, IL) could be used in future studies to create a 3D reconstruction of the spine and trunk shape instead of the x-ray’s 2D depiction [45]. Surface topography has been used as a radiation-free assessment tool to assess the posterior deformity and changes associated with scoliosis over time [45,46,47]. This would help to determine how strong the correlation is between body image disturbances and objective trunk shape analysis in AIS females.
- I feel that the conclusion does not reflect the purpose of the study. It is too general. Your study is a picture of “self-esteem” and internalization of AIS patients at a moment of their evolution. It does not bring the evidence that “internalization” modifies the results or the perception of the results of bracing/surgery. Conclusion should focus on the importance of screening high “internalized” patients that could possibly help us understanding why a group of patients is pushing for treatment (surgery) based on aesthetical concerns.
We agree with the Reviewer’s opinion that the conclusions do not reflect our study aims. They are, in fact, too general. We have reformulated the conclusions as follows:
Conclusions
The presented study has shown that COPS and SATAQ-3 are tools with satisfactory reliability. It confirmed the usefulness of those questionnaires in scientific research and adolescent screening. This procedure could indicate patients scoring high on COPS and SATAQ-3, who are strongly determined to undergo AIS scoliosis treatment mostly due to aesthetical concerns. Moreover, further analyses and observations on use of COPS and SATAQ-3 could also be conducted on various, clinical and general, populations of Poles to enrich this tool with the development of norms for individual categories.
Reviewer 2 Report
Review
Introduction
I would better describe what is “internalization” in the intro.
Please shorten the SATAQ and COPS description in the intro. Part of it could be transferred to the discussion.
L101 remove « by patients”
M&M
Since you refer in the results to different items of each questionnaire, I feel it is important to place the Questionnaires in M&M.
Please clarify the method with sub-sections.
It is really difficult to understand the results without a better description of what was measured and computed.
Results
Please explain “retest” in the tables.
Have sent the questionnaire twice ? If so please revise M&M.
Tables are mostly unreadable since it is not possible to understand each item signification.
Discussion
The first 2 paragraphs repeat the intro. I recommend focusing on the study results.
One of the main limitations of the study is that there was no objective trunk shape analysis.
Rib hump magnitude for instance is not well correlated to radiological features, which may explain the poor correlation between cosmetic and radiological features.
3D surface topography analysis should be discussed. There is a high correlation between self-image and trunk shape.
I feel that the conclusion does not reflect the purpose of the study. It is too general. Your study is a picture of “self-esteem” and internalization of AIS patients at a moment of their evolution. It does not bring the evidence that “internalization” modifies the results or the perception of the results of bracing/surgery.
Conclusion should focus on the importance of screening high “internalized” patients that could possibly help us understanding why a group of patients is pushing for treatment (surgery) based on aesthetical concerns.
Author Response

(The authors gave the same response as above.)

Round 2
Reviewer 1 Report
Thanks and Regards.
Author Response
Thank You for your time and support. The manuscript has been changed according to Your and Editor's suggestions.
Best Regards